



# tobac v1.0: towards a flexible framework for tracking and analysis of clouds in diverse datasets

Max Heikenfeld[1], Peter J. Marinescu[2], Matthew Christensen[1], Duncan Watson-Parris[1], Fabian Senf[3], Susan C. van den Heever[2], and Philip Stier[1]

[1]Atmospheric, Oceanic & Planetary Physics, Department of Physics, University of Oxford, Oxford, United Kingdom
[2]Department of Atmospheric Sciences, Colorado State University, Fort Collins, USA
[3]Leibniz Institute for Tropospheric Research, Leipzig, Germany

**Correspondence:** Max Heikenfeld (max.heikenfeld@physics.ox.ac.uk)

**Abstract.** We introduce tobac (Tracking and Object-Based Analysis of Clouds), a newly developed framework for tracking and analysing individual clouds in different types of datasets, such as cloud-resolving model simulations and geostationary satellite retrievals. The software has been designed to be used flexibly with any two- or three-dimensional time-varying input. The application of high-level data formats, such as iris cubes or xarray arrays, for input and output allows for convenient use

of metadata in the tracking analysis and visualisation. Comprehensive analysis routines are provided to derive properties like cloud lifetimes or statistics of cloud properties along with tools to visualise the results in a convenient way. The application of tobac is presented in two examples. We first track and analyse scattered deep convective cells based on maximum vertical velocity and the three-dimensional condensate mixing ratio field in cloud-resolving model simulations. We also investigate the performance of the tracking algorithm for different choices of time resolution of the model output. In the second application, we

show how the framework can be used to effectively combine information from two different types of datasets by simultaneously tracking convective clouds in model simulations and in geostationary satellite images based on outgoing longwave radiation. tobac provides a flexible new way to include the evolution of the characteristics of individual clouds in a range of important analyses like model intercomparison studies or model assessment based on observational data.

## 1 Introduction

Clouds are a major feature of the Earth's atmosphere and control many critical processes in the Earth's energy and water budgets (Trenberth et al., 2009). Different types of convective clouds play important but distinct roles in many regions of the globe. Shallow cumulus clouds are widespread over the subtropical trade-wind latitudes and have a strong impact on the radiative balance of the atmosphere, including a potential for strong feedbacks from anthropogenic perturbations of the climate system (Stevens and Feingold, 2009). Deep convective clouds are a defining element of the atmosphere over most of the tropics

(Nesbitt et al., 2006), driving both local weather dynamics and larger scale circulation patterns, which has impacts on the entire climate system (Emanuel, 1994). Furthermore, deep convective clouds play a major role in extreme weather events all over the globe (Doswell, 2001; Gensini and Mote, 2014). Therefore, clouds and their interactions with other aspects of the climate system are an essential aspect of many important challenges in our understanding of the Earth's atmosphere and current changes





due to anthropogenic influences (IPCC, 2013). The nature of convective clouds is highly localised. Individual convective cells undergo rapid dynamic development over relatively short timescales of minutes to hours (Orlanski, 1975), while organised convective features, such as mesoscale convective systems (MCS), can persist for many hours or even days (Orlanski, 1975; Laing and Fritsch, 1997; Fritsch and Forbes, 2001; Feng et al., 2018). Further advances in understanding the physical processes

underlying the development of these clouds require analysis techniques that go beyond the usual approaches, which are often based on bulk statistical properties over larger regions in space and time, such as entire modelling or observational domains. Model intercomparison studies with cloud-resolving model (CRM) simulations have mostly relied on the comparison of domain and time-averaged quantities or similar statistics (Varble et al., 2011, 2014a, b; Fan et al., 2017). This generally limits the investigation of differences between the models on the scale of individual convective cells or analyses that take the temporal

evolution of individual clouds into account.

Any analysis that focuses on the properties of individual clouds in larger databases containing numerous cloud elements, and aims at including the time evolution over their development cycle, requires some form of cloud tracking technique. A large body of work exists on tracking individual clouds in different types of data, ranging from ground-based radar and geostationary satellite retrievals to model simulations at a range of different resolutions. We now present a short but certainly not exhaustive

overview of existing approaches. This will be used to show the capabilities of the existing software and to discuss drawbacks or limitations which motivated the development of the more flexible software framework tobac (Tracking and Object-Based Analysis of Clouds) presented here.

Tracking of individual convective clouds in radar data has been performed for decades (Crane, 1979; Rosenfeld, 1987). These efforts were often motivated by their use in nowcasting of severe weather warnings, e.g. for flooding due to convective precipi-

tation, damage from hail or impacts of high wind speeds such as tornadoes (Dixon and Wiener, 1993; Lakshmanan and Smith, 2009). Satellite-based tracking of convective clouds has been performed both with a similar focus on nowcasting convection and for long-term analysis in climate research (Menzel, 2001; Sieglaff et al., 2012). Special tracking algorithms that combine information from different wavelength bands of imagers on geostationary satellites, such as Cb-TRAM (Zinner et al., 2008, 2013) and RTD (Autonès and Moisselin, 2013) have been developed as tools to identify and track deep convective clouds

throughout their development cycle, including the initial stage of rising cumulus towers. However, both products have been developed for a specific application that strongly limits a more general adaption of the software by the user. Several other studies have used geostationary satellite data to investigate the growing phase and glaciation of deep convective clouds (Mecikalski and Bedka, 2006; Mecikalski et al., 2011; Senf et al., 2015; Senf and Deneke, 2017). Other applications have specifically focused on the analysis of long-lived MCSs over different regions of the globe (Machado et al., 1998; Feng et al., 2012; Hagos

et al., 2013; Feng et al., 2018).

Tracking of individual cloud objects in high-resolution CRM simulations and large-eddy simulation models (LES) has been developed alongside the evolution of these simulations in recent decades. Earlier studies on tracking shallow convection in high-resolution model simulations (Zhao and Austin, 2005a, b; Heus et al., 2009) strongly relied on manual detection techniques. Subsequently, Dawe and Austin (2012) and Heus and Seifert (2013) presented automated methods of tracking shallow

convection that rely on a continuous release of a decaying tracer from the model as described in Couvreux et al. (2010). How-



ever, the functionality of the tracer release and advection must be specifically implemented in each model and restricts the use of this technique to output of high-resolution models. Cloud tracking algorithms applied online during the actual model simulations (Plant, 2009) have the advantage of direct access to the relevant model fields at the model timestep and thus the highest possible time resolution. However, these online algorithms must also be implemented separately in a specific model.

Moseley et al. (2013, 2016) tracked precipitation patterns for investigations of deep convective clouds and convective invigoration. Davis et al. (2006, 2009) presented an object-based analysis of rainfall patches, including tracking capabilities, which was applied to precipitation on a relatively large regional scale. Heiblum et al. (2016a, b) developed and applied a tracking algorithm for warm convective clouds that determines cloud volumes from the condensate mixing ratio field and then propagates the clouds based on the velocity of the cloud centre of mass. This algorithm allows for cloud splits and merges to form com-

plex cloud entities possibly involving numerous individual clouds. Only a few studies have focused on tracking individual deep convective clouds in model simulations in a way that takes into account the actual cloud volumes (Chen et al., 2017). Terwey and Rozoff (2014) developed a tracking algorithm for individual convective updrafts and applied it to CRM simulations of hurricane cases with two different models. However, this effort has not been provided to the community as a generalised software package aimed at more widespread use cases. Several other approaches that included the tracking of individual updrafts in

different types of cumulus clouds in a very detailed manner (Sherwood et al., 2013; Hernandez-Deckers and Sherwood, 2016) would not be easily transferable to data with a lower temporal and spatial resolution.

Despite these advances in developing detailed cloud tracking approaches for use in highly resolved model simulations, most current studies are performed with model grid spacings of several hundreds of metres to a few kilometres, especially when using larger domains or simulations for longer time periods. Providing adequate ways of performing tracking and object-based

analyses for different types of clouds, including deep convection, in these kinds of simulations provides a key pathway to better understanding the underlying physical processes.

This overview clearly shows the wide range of extensive efforts that went into the development of elaborate software and analysis tools to track clouds in different types of datasets. However, it also highlights the problem of limited compatibility between the different existing approaches and implementations, especially regarding the intended use of tracking clouds based

on different data sources using the same algorithms and analysis tools.

To address some of these limitations of existing approaches and provide a more functional tool with increased flexibility for different applications, we have developed tobac as a new flexible software tool for the identification, tracking and analysis of clouds. This approach certainly does not intend to replace the existing tools in their specific applications, but is rather aims to provide a flexible framework that can be used for a wide range of different datasets and also allows the future integration of

some of the existing approaches discussed here.

tobac is designed in a modular way that includes the following basic steps, which are described in detail in the following sections 2.1–2.5:

1. Data input and output

2. Feature detection





3. Segmentation of cloud areas/volumes

4. Trajectory linking

5. Object-based analysis and visualisation

tobac provides a framework that allows for a convenient application to output from a wide range of model simulations and
observational products, as long as it is provided with sufficient temporal and spatial resolution and contains output variables
that can be used to identify individual clouds. Therefore, the software package can be used for a range of important applications
like model intercomparison studies, which generally rely on simpler analysis methods that do not capture the evolution of indi-
vidual clouds. These capabilities also allow for comparative studies between model simulations and observational datasets, e.g.
from satellite retrievals, using the same underlying statistical methods. Due to the modular structure, tobac is set up for the in-
tegration of existing or newly developed algorithms for the different steps in the analysis chain. The implementation in Python
provides tobac with access to numerous more specialised existing software libraries for different aspects of the software, such
as data input/output, memory usage and the existing functionality from the field of image processing. We also show how we
can leverage an existing Python library from an entirely different field of the physical sciences to perform integral parts of the
linking step in our application. Furthermore, the choice of Python for tobac makes the package more easily accessible to users
as it allows for easier integration into existing analysis workflows and also stimulates the integration of additional components
in the modular workflow of the package.

To show the advantages of tobac in practical applications, we present two different examples of using the framework in tracking
and analysing deep convective clouds. In the first application, the detection of features is performed on the column-maximum
vertical velocity at each output time interval from a CRM simulation. A three-dimensional watershedding algorithm is applied
to the updraft field and to the total condensed water content field (mass mixing ratio of all hydrometeors) at each step in time
to infer both the volume of the individual updrafts and the clouds associated with the tracked updrafts. These features are
then filtered and linked into consistent trajectories. We use the tracking results to assess the distribution of cloud lifetimes and
the requirements for the model output temporal resolution. In the second application, we perform a simultaneous analysis for
model and satellite data. Similar vertically resolved data as used in the other example are usually not available from satellite
imagers. The information in most satellite retrievals of cloud properties is limited to two-dimensions. With a multi-spectral se-
lection of channels from the satellite instrument, cloud-top height and radiative properties can be retrieved (McGarragh et al.,
2018). An analysis of model-simulated and satellite-retrieved fields of outgoing longwave radiation is presented to demonstrate
the flexibility of the tobac approach. By making use of the framework consistently across different datasets like this, we can
compare the tracked clouds in both data sources by examining the statistical properties of the resulting population of convective
clouds, thereby facilitating model-observation comparisons.



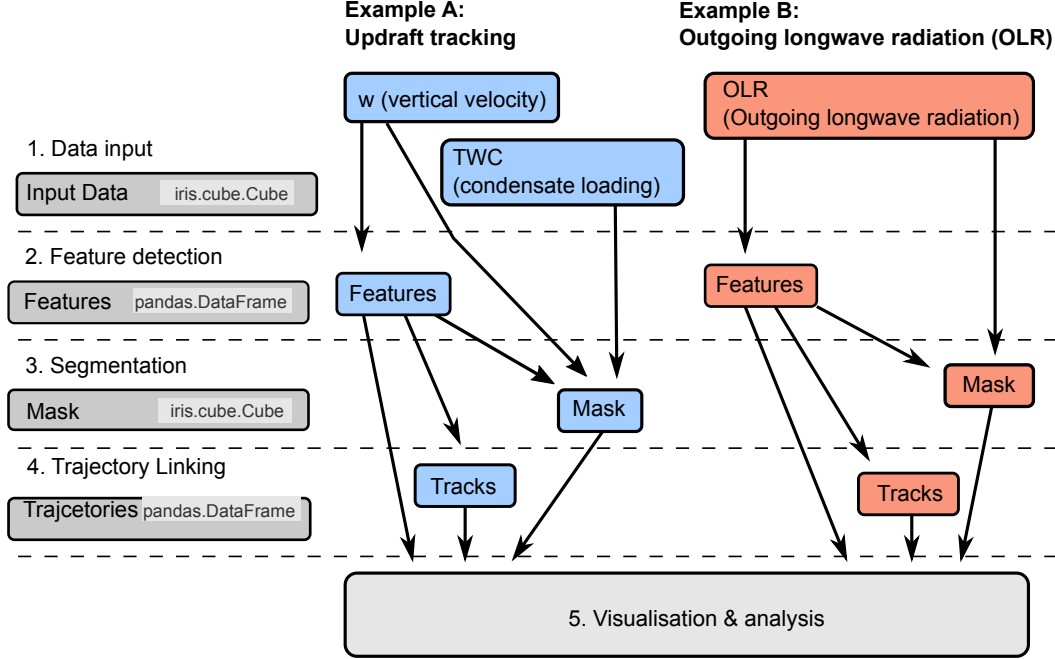

**Figure 1.** Schematic overview of the general workflow of the tobac tracking analysis framework and of the two examples presented in this manuscript.

## 2   Software description

In this section, we describe the general design and workflow of the software package as illustrated in Fig. 1 for the two example applications presented in Sect. 3 and Sect. 4. The implementation of the individual analysis steps described here reflects an example combination of analysis steps currently implemented in tobac. Due to the modular setup of the package, different parts

5   of the workflow can be combined in a different way or replaced by future additions to the framework.

### 2.1   Data input and output

The input data for the framework are provided in a high-level format of either Iris cubes (Met Office, 2018) or xarray DataArrays (Hoyer and Hamman, 2017), which include detailed metadata for each data variable, such as units and coordinates. The algorithm can thus automatically use these metadata, and the tracking setup can be controlled independently of the tempo-

10   ral resolution, spatial resolution or dimensions of the input data. Tracking parameters representing physical properties like distances or time periods can be set in physical units and are automatically converted to pixel-based values needed for the underlying calculations. Scientific data are provided in a vast variety of different file formats and data structures. Implementing a way of loading the data into the right format for an application often proves to be a significant limitation to the use of new datasets and generally consumes an unjustifiable amount of time and effort, apart from providing a significant source of



implementation errors. The Python library Community intercomparison Suite (CIS) (Watson-Parris et al., 2016) overcomes this challenge and provides a convenient way to automatically load a vast array of observational datasets into Iris-compatible objects for direct use in tobac.

Both Iris and xarray make use of so-called "lazy loading" based on the dask package (Rocklin, 2015; Dask Development Team,
2016) for efficient memory usage. The initial loading of data from a file only creates a place-holder. Then, individual operations on the data are combined and evaluated once the final result is to be saved, printed or plotted. Only at this stage are data actually loaded from disk into the physical memory of the computing machine and individual calculations performed. Based on these capabilities, the entire tobac framework is written with a focus on limiting instantaneous memory usage by splitting up calculations into chunks, e.g. along the time dimension. Hence, even large datasets with individual fields much larger than
the memory available on the computing system can be conveniently processed without special adaptation by the user.

The output of the tracking analysis is given in commonly used high-level Python data format as pandas DataFrames (McKinney, 2010) for a table containing the tracked cell centres and trajectories and as Iris cubes or xarray DataArrays for the masks of cloud volumes or areas. This output is automatically amended with the same metadata as the input data like coordinates
(e.g. time, longitude, latitude), along with additional information from the tracking process, e.g. a time coordinate relative to the initiation of an individual convective cell. This allows for the convenient and direct use of the output for visualisation and further analyses.

## 2.2 Feature detection

The feature detection can work with any two-dimensional field either present or derived from the input data. In the first example, we use maxima in the maximum vertical velocity in each column of the three-dimensional high-resolution model output to identify individual updrafts (see Sect. 3). In the second example, minima in outgoing longwave radiation from satellite retrievals and model output are used in the feature detection (see Sect. 4).

To identify the features, contiguous regions above or below a threshold are determined and labelled individually. Smoothing
the input data, e.g. with a Gaussian filter, makes this step much more reliable. Erosion of the labelled regions by a specified length or number of pixels leads to more robust detection of individual features as described in Senf et al. (2018).

To describe the specific location of the feature at a specific point in time, we have investigated the use of different spatial properties describing the identified region. The geometric centre can be strongly affected by changes in the shape of the boundary, which is determined based on the selected threshold value. Instead, we have found that a weighted mean

$$\boldsymbol{x}_{\mathrm{mean}} = \sum_i w_i \boldsymbol{x}_i \tag{1}$$



with weights $w_i$ given by the difference between the values of the chosen field at the individual points $V_i$ and the threshold value $V_{feature}$

$$w_i = \frac{V_i - V_{feature}}{\sum_i V_i - V_{feature}} \tag{2}$$

has proven to perform best in determining a robust feature position. We can interpret this position as the centre of mass of the
component of the field exceeding the chosen threshold value.

Using a single threshold to identify features can lead to problematic results in two different ways. A very restrictive threshold means that clouds with weak vertical velocities or clouds during their initial and decaying stage will not be captured. On the other hand, a weakly restrictive threshold can lead to spurious results as it might lead to large unconfined regions around deep convection being selected, or to an unwanted merging of several distinct cloud features into one. To resolve these conflicting
requirements arising in the case of a single threshold value, we have developed a step-wise approach with a range of threshold values (Fig. 2). These threshold values have to be chosen specifically for each application of tobac. The choice can be based on a detailed analysis of the data used for tracking to determine where the features separate from the background, e.g. based on histograms as shown in Sect. 4, or using empirical values from previous studies of the specific phenomena. The feature identification starts with labelling the regions for the least restrictive threshold. For each threshold value, features are identified
in the same way (Fig. 2b, d, f) and replace existing features that were found based on a less restrictive threshold value in the surrounding region (Fig. 2e, g).

This combination of different thresholds allows tobac to detect lower intensity features representing weaker convective clouds or clouds in their initial or decay stage but identify localised features with stronger updrafts or colder cloud tops within the weaker-threshold areas. In the first example application (Sect. 3), consecutive maximum updraft threshold values of 3, 5 and
$10\,\mathrm{m\,s^{-1}}$ were used, while tracking based on OLR in the second example (Sect. 4) was performed with consecutively smaller threshold values (250, 225, 200, 175 and 150 $\mathrm{W\,m^{-2}}$).

## 2.3   Segmentation

Once features and feature centres are identified, segmentation techniques are used to associate areas or volumes to each identified feature. In the current version of the tobac framework, we have implemented segmentation using watershedding techniques
from the field of image processing (skimage.morphology.watershed from the scikit-image library (Soille and Ansoult, 1990; van der Walt et al., 2014) with a fixed threshold value $V_{seg}$. This value has to be set specifically for every type of input data and application, as explained in more detail for the two example applications in Sect. 3 and Sect. 4. Watershedding is widely used in several existing cloud tracking and analysis algorithms described in Sect. 1, such as Heiblum et al. (2016a), Fiolleau and Roca (2013) and Senf et al. (2018).
This segmentation routine can be performed for both two-dimensional and three-dimensional data. At each timestep, a marker is set at the position (weighted mean centre) of each feature identified in the detection step in an array otherwise filled with zeros. In the case of the three-dimensional watershedding, all cells in the column above the weighted mean centre position of the identified features fulfilling the threshold condition $V_{segmentation}$ are set to the respective marker. The algorithm then fills the



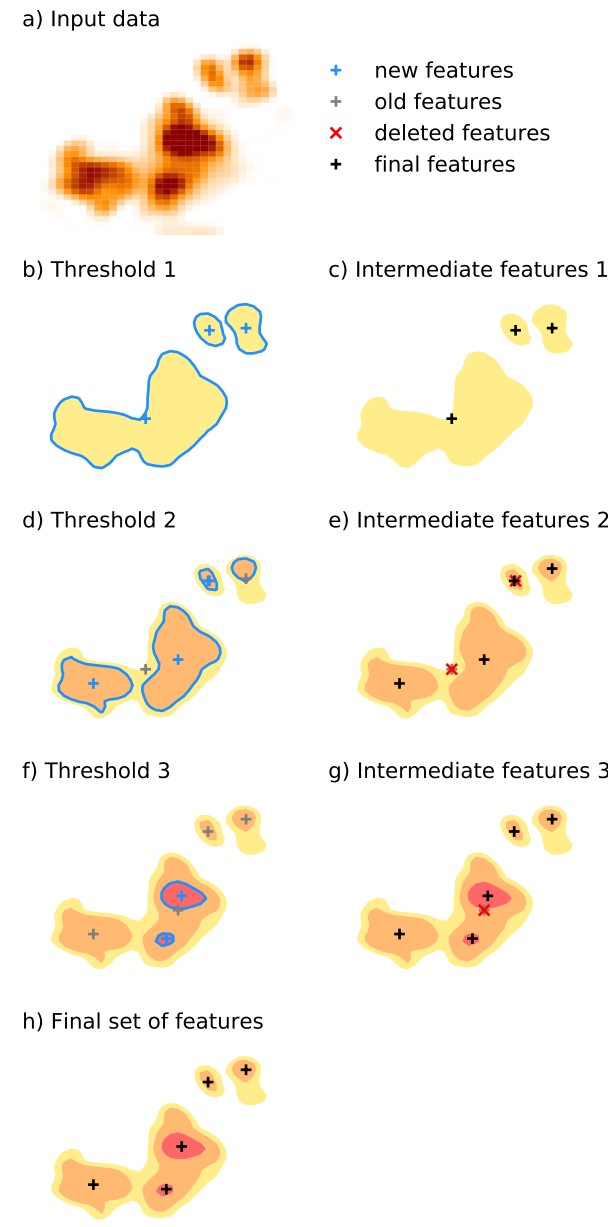

**Figure 2.** Schematic illustration of the multi-threshold feature detection approach using three different threshold values.





area (2D) or volume (3D) based on the input field starting from these markers until reaching the threshold $V_{\text{segmentation}}$. If two or more cloud objects are directly connected, the border runs along the watershed line between the two regions. This procedure creates a mask of the same shape as the input data, with zeros at all grid points where there is no cloud or updraft and the integer number of the associated feature at all gridpoints belonging to that specific cloud/updraft. This mask can be conveniently and efficiently used to select the volume of each cloud object at a specific timestep for further analysis or visualisation.

The structure of tobac allows for the future implementation of other algorithms for the segmentation step, e.g. replacing the watershedding approach by random walk techniques (Grady, 2006; Wang et al., 2019) or other image processing tools.

## 2.4 Trajectory linking

The individual features and associated area/volumes identified in each timestep have to be linked into cloud trajectories to analyse the time evolution of cloud properties for a better understanding of the underlying physical processes. For this step, we have implemented a linking method that makes use of trackpy (Allan et al., 2016), a Python library originally developed for tracking particles and cells in microscopic images. The linking determines which of the features detected in a specific timestep (see Sect. 2.2) is identical to an existing feature in the previous timestep. For each existing feature, the movement within a time step is extrapolated based on the velocities in a number of previous time steps. The algorithm then breaks the search process down to a few candidate features by restricting the search to a circular search region centred around the predicted position of the feature in the next time step. For newly initialised trajectories, where no velocity from previous timesteps is available, the algorithm resorts to the average velocity of the nearest tracked objects.

The parameter $v_{\text{max}}$ restricts how much the future position of a feature is allowed to deviate from a linear extrapolation of the trajectory over time. It thus has the units of a velocity and describes the dependency of the circular search range $d$ on the output timestep $\Delta t$ in the data used for the tracking

$$d = v_{\text{max}}\Delta t. \tag{3}$$

In the applications (Sect. 3 and 4), we set this value to $v_{\text{max}}=10\,\text{m}\,\text{s}^{-1}$, which results in a search range of $600\,\text{m}$ around the projected position for 1 minute data input and $3\,\text{km}$ for 5 minute data input. Variations in the shape of the regions used to determine the positions of the features can lead to quasi-instantaneous shifts of the position of the feature by one or two grid cells even for a very high temporal resolution of the input data, potentially jeopardising the tracking procedure. To prevent this, tobac uses an additional minimum radius of the search range $d_{\text{min}}$ ($2\,\text{km}$, equivalent to four times the grid spacing in Sect. 3) that specifies a lower limit for the size of the search region. Both these parameters are given as physical quantities and then converted into pixel-based values used in trackpy. This allows for cloud tracking that is controlled by physically-based parameters that are independent of the temporal and spatial resolution of the input data. We make use of this for cloud tracking with different model output frequencies for the same simulations in the example application in Sect. 3.

Features can be allowed to be missed for a certain number of timesteps (*memory*) and still get linked into a trajectory. However, this option should be used with caution, as it can lead to erroneous trajectory linking, especially for data with low time



resolution. For example, convective clouds can produce outflow boundaries that initiate new convective clouds nearby, and the newly-formed clouds are more likely to be linked to the original clouds with this option. The trajectories can also be extrapolated to additional output timesteps at the start and at the end of the tracked path. This allows for the inclusion of both the initiation of the cell and the decaying later stages of the cell development that may have been unidentified based on the chosen

thresholds. Furthermore, a threshold for the minimum lifetime of the tracked objects can be used to exclude the analysis of clouds that have only been tracked for a very short period and are likely to be spurious features. Such tracked objects can contaminate analyses focusing on the cloud lifetime and associated quantities.

The trajectories are recorded in a pandas DataFrames. This enables filtering the resulting trajectories, e.g. to reject trajectories that are only partially captured at the boundaries of the input field both in space and time.

The current version of the linking step does not include an explicit treatment of the splitting and merging of clouds, as implemented in several of the cloud tracking algorithms reviewed earlier (Dawe and Austin, 2012; Heus and Seifert, 2013; Heiblum et al., 2016a). Instead, the algorithm creates a continuous track for the cloud that most directly follows the direction of travel of the preceding or following cell path. However, we have structured the implementation of tobac in a way that allows for the future addition of more complex tracking methods recording a more complex network of relationships between cloud objects

at different points in time.

## 2.5   Object-based analysis and visualisation

To make use of the results of the previous steps, we provide detailed tools to analyse and visualise the tracked objects. We provide a set of routines that enable performing analyses and deriving statistics for individual clouds, such as the time series of integrated properties and vertical profiles. We also provide routines to calculate summary statistics of the entire population

of tracked clouds in the cloud field like histograms of cloud areas/volumes or cloud mass and a detailed cell lifetime analysis (see Fig. 5 and Fig. 9).

These analysis routines are all built in a modular manner. Thus, users can reuse the most basic methods for interacting with the data structure of the package in their own analysis procedures in Python. This includes functions performing simple tasks like looping over all identified objects or cloud trajectories and masking arrays for the analysis of individual cloud objects. Plotting

routines include both visualisations of the entire cloud field and detailed visualisations for individual convective cells and their properties.

## 2.6   Advantages of the implementation in Python

While the majority of the existing tracking approaches reviewed in Sect. 1 are implemented either in Fortran, C and C++ or in proprietary programming languages like MATLAB, we have chosen to use Python for our tracking framework for several

practical reasons. Python has become the go-to standard for data analysis in many fields of scientific research, including the atmospheric sciences in recent years (Lin, 2012; Perkel, 2015). This makes it possible to develop software that is accessible and modular, which allows for the successful addition of user-contributed algorithms or the adoption or application of the workflow for cases beyond those presented here. The use of libraries in the scientific Python ecosystem including NumPy, SciPy, and





matplotlib (Hunter, 2007; van der Walt et al., 2011), along with a large stack of existing and optimised libraries providing image processing features (van der Walt et al., 2014), means that the package is based on actively developed open-source projects. This ensures an accurate, effective and tested implementation of the individual calculations as well as the continuous integration of new developments and improvements. Most of these Python libraries use Fortran or C for the actual underlying

calculations, which means that many of the individual operations within tobac make use of the increased computational speed of these languages. The use of Python also means that even users without extensive programming experience will be able to easily adapt existing procedures into the workflow or contribute additional algorithms to the modular structure of the tobac tracking framework.

The implementation in Python also enables the use of Jupyter notebooks (Perez and Granger, 2007; Kluyver et al., 2016) as an

innovative way of developing, visualising and sharing scientific data analyses. We provide examples of the analyses presented here as Jupyter notebooks provided in the software package.

Memory limitations have been cited as a significant challenge for the application of many of the presented algorithms (Dawe and Austin, 2012; Heus and Seifert, 2013). The use of modern memory management techniques such as "lazy data loading" in the underlying Python libraries Iris (Met Office, 2018) or xarray (Hoyer and Hamman, 2017), which both rely on the dask data

types (Rocklin, 2015), allows for clear and concise source code while sparing the users of having to deal with memory-related considerations themselves.

## 3   Example A: tracking of convective cells in high-resolution model simulations based on updraft velocities and condensate mixing ratios

In the first example, we apply the tracking framework to CRM simulations of scattered deep convection. Deep convective

clouds are characterised by regions of strong vertical motions which are concentrated in relatively confined updraft cores that dominate the dynamics of the cloud evolution (Cotton et al., 2010). Hence, the updraft cores are well suited to be used for identifying and tracking individual convective cells. We use the total condensate mixing ratio, i.e. the total amount of liquid and frozen water per mass of dry air, to associate the identified updraft cores with the respective cloud volume at each time. We make use of simulations that were performed as part of a larger model intercomparison case study in the deep convection

working group of the Aerosol, Clouds and Precipitation (ACPC) initiative (van den Heever et al., 2017) aimed at understanding the response of scattered convection to changes in aerosol number concentrations. The tracking algorithm presented here will be used as part of the analysis for the model intercomparison study using several different three-dimensional CRMs. In this manuscript, we show the identification and tracking of deep convective clouds on a model grid with horizontal grid spacing of $500\,\mathrm{m}$ and vertical grid spacing of $50$–$300\,\mathrm{mm}$ for simulations with the Weather Research and Forecasting (WRF) model

(Skamarock et al., 2005). The simulations use the Morrison microphysics scheme (Morrison et al., 2005, 2009) and the Rapid Radiative Transfer Model (RRTMG) short and longwave radiation scheme (Iacono et al., 2008). Initial and boundary conditions for the simulations in a nested setup with two additional grids of coarser grid-spacing ($1.5\,\mathrm{km}$ and $4.5\,\mathrm{km}$) around this target domain were provided by the GDAS-FNL reanalysis National Centers for Environmental Prediction (2015). The simulation



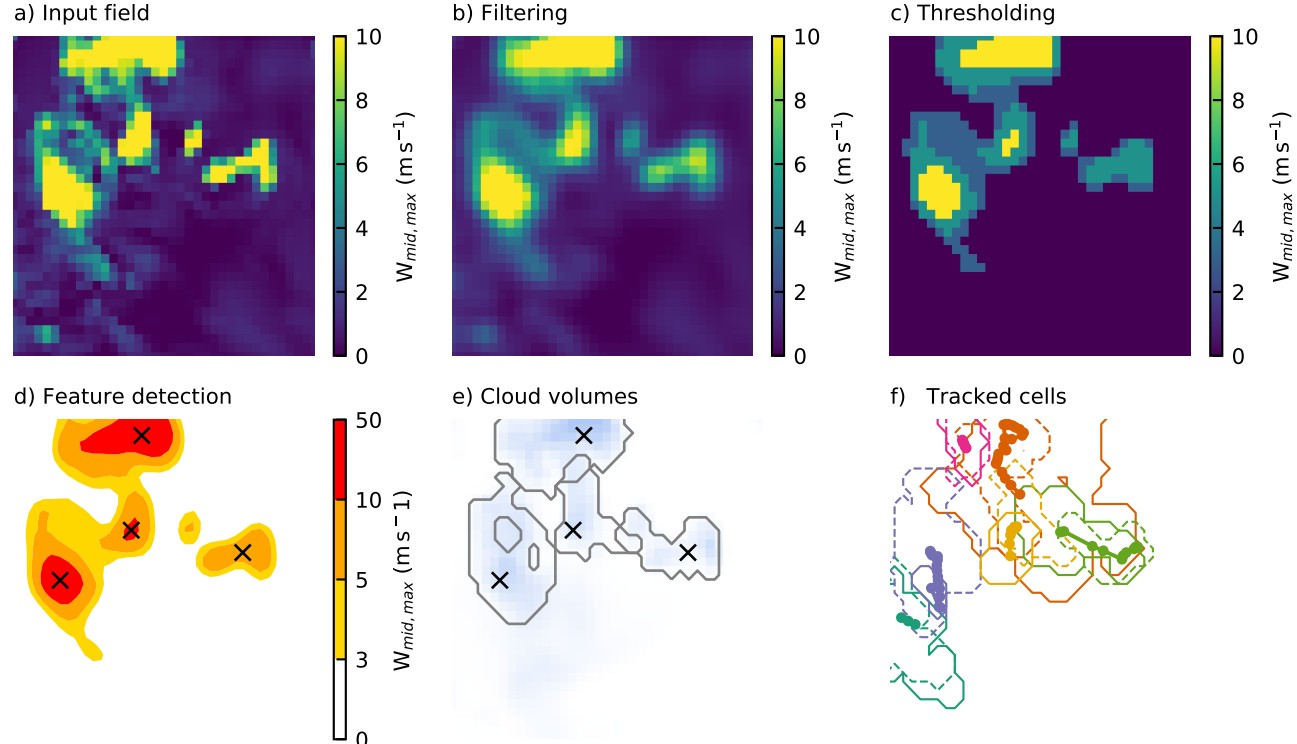

**Figure 3.** Schematic overview of the individual steps of the tracking algorithm for an example subset of the domain used in example A including the input mid-tropospheric velocity field. The input data (a) are smoothed with a filter (b) before regions above or below a set of thresholds are determined (c) to identify the individual features (d). (e) shows the surface projection of the associated cloud volumes determined in the segmentation set and (f) shows the entire trajectories of the cells present at this timestep, including the surface projection of the cell volume at the start (dashed) and at the end (solid) of the trajectory.

results are output at a frequency of 1 minute for an extended part of the simulation period (3 hours) and at a frequency of 5 minutes for 12 hours of the simulations. We use a combination of the three-dimensional fields of vertical velocity and total condensate mixing ratio in this application to track individual convective clouds.

5    The individual steps of the analysis are visualised for a specific point in time and a subset of the model domain in Fig. 3. The three-dimensional vertical velocity field is reduced to the maximum updraft velocity in each model column over a mid-tropospheric range of geopotential height ($3000\,\mathrm{m}$ to $8000\,\mathrm{m}$) (Fig. 3a). This avoids the impact of strong vertical motions both in the lower troposphere, that may be associated with outflow boundaries, and also gravity waves in the upper troposphere. A Gaussian filter with a variance of $\sigma = 1\,\mathrm{km}$ is used to filter the input in the feature detection step (Fig. 3b) to create a smoother

10   field that assists in the feature detection. This two-dimensional field is then used to identify individual deep convective updrafts in the simulation. The feature identification following Sect. 2.2 is performed with a set of three updraft velocity thresholds





of $3\,\mathrm{m\,s^{-1}}$, $5\,\mathrm{m\,s^{-1}}$ and $10\,\mathrm{m\,s^{-1}}$ (Fig. 3c) and yields the individual features marked in Fig. 3d. Segmentation is performed on the condensate mixing ratio using the watershedding technique (see Sect. 2.3) with a threshold of $0.5\,\mathrm{g\,kg^{-1}}$ to identify the cloud volumes corresponding to the individual identified updrafts. The cloud volumes derived with watershedding from the condensate mixing ratio field of each of the identified updrafts is represented by the surface projection of the 3D volumes

(Fig. 3e). Note that the intersecting lines in Fig. 3e represent instances where cloud volumes associated with different updraft cores may be present in the same column but at different altitudes. Trajectories are formed by linking up the individual features and are shown including the surface projection of the cloud volumes at the initial and final timestep of each tracked cell (Fig. 3f). A smaller subset of the data and analysis for this example including the tracking analysis and visualisation is available as a Jupyter notebook as part of the package source code.

## 3.1   Time resolution requirements for cloud tracking

The cloud tracking framework presented here can be applied to model output from any atmospheric model simulation with sufficient resolution to resolve the features intended to be studied. However, successful tracking of individual clouds in the simulation output requires sufficiently high spatial and temporal resolution. However, writing output data at high frequency from numerical model simulations drastically increases the computational expense of the simulations and the size of the output

datasets. For observational data, such as geostationary satellite data, the available time resolution might be limited by technical restrictions such as scanning time or data transmission. It is thus important to determine the necessary input frequency for the successful tracking of a specific type of dataset.

The tracking step (Sect. 2.4) uses trackpy, which is based on the tracking methods developed for in Crocker and Grier (1996). The algorithm has been originally developed for microscopic particles; however, all considerations apply equally to the tracked

features we regard here in tobac. In their development of the algorithm, the authors state that successfully linking objects into trajectories is only feasible if the typical displacement of a particle during one time step is significantly smaller than the typical inter-particle spacing. To assess how valid these assumptions are for our application, we investigated the nearest-neighbour distances for individual cells and the typical displacement of the tracked objects within one timestep. Distances between cloudy updrafts (Fig. 4a) were most frequently around $5\,\mathrm{km}$ with a significant tail of up to $30\,\mathrm{km}$ representing more isolated cells.

This distribution is independent of the chosen output timestep as it represents an instantaneous relationship between cells at individual points in time. The updraft propagation velocities derived for tracking with a 1 minute output timestep (Fig. 4b) were most frequently at around $4\,\mathrm{m\,s^{-1}}$ with more than $90\,\%$ of the velocities below $10\,\mathrm{m\,s^{-1}}$.

Using the output timestep and these velocities, we can calculate the displacement of the clouds within one tracking timestep and compare that to the nearest-neighbour distances (Fig 4c). In addition to the timestep of 1 minute, the displacements that

would result from lower output frequencies of 5, 10, 15 and 30 minutes based on these velocities were calculated (Fig 4c). While there is no significant overlap between the nearest-neighbour distance distribution and the displacement distribution for an output timestep of 1 minute, the tails of the distributions start overlapping for 5-minute input data, although the peaks are still distinctly separate. For lower output frequencies of 15 minutes and 30 minutes, however, there is a clear overlap between the nearest-neighbour distance distribution and the distributions of displacement within one time step. Therefore, these fre-





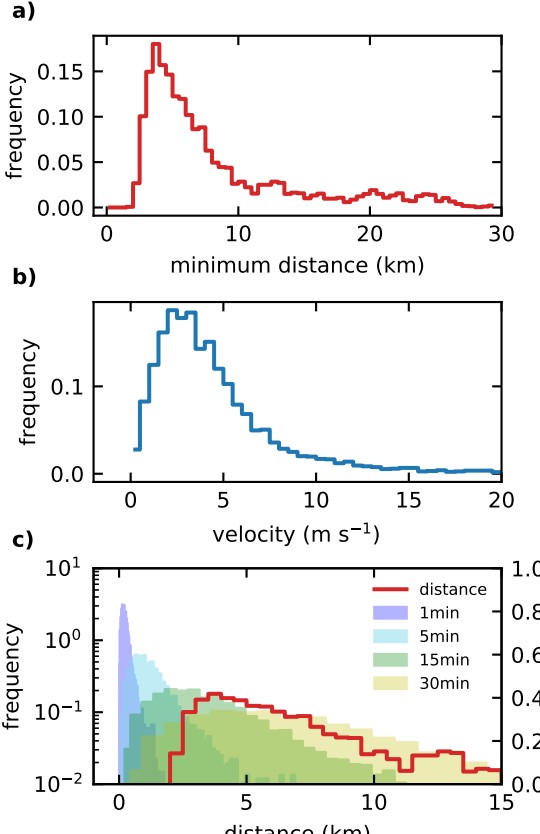

**Figure 4.** (a) Distributions of the distance to the next identified object for all identified objects and b) velocities for tracked cloud objects at each timestep of the trajectories. (c) The distribution of derived travel distances of individual clouds during one output timestep (shaded colours) resulting from these velocities in (b) is shown together with the distribution of the minimal distance to the nearest-neighbour for individual objects as shown in (a).

quencies would be outside the range postulated for the successful application of the tracking algorithm used by trackpy. Hence, when applying this tracking algorithm, it is important to understand both the spatial distribution of the desired tracked features and their propagation velocities to ensure that the output timestep is sufficiently frequent. For the simulations assessed here, both 1-minute and 5-minute output frequencies would be acceptable for tracking cloudy updrafts, with 1-minute output likely

5 to provide more successful and accurate tracks.

The cloud lifetimes (Fig. 5) are analysed for the same 3-hour period using the two different time resolutions (1 minute and 5 minutes) and agree well for clouds with lifetimes larger than about 15 to 20 minutes. For shorter lifetimes, the 1-minute input data yield significantly more tracked cells. It is obvious that we can only properly represent and analyse cloud lifetimes for clouds that exist over a certain number of output timesteps in this framework. An individual cloud that is tracked for 5 to 10





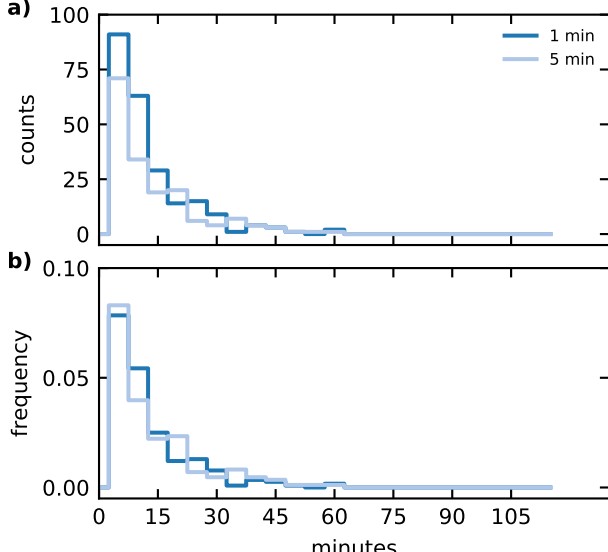

**Figure 5.** Cell lifetimes for tracking and analysis using two different output timesteps (1 minute and 5 minutes) showing both total counts (a) and the PDF (b)

minutes based on 1-minute output allows for robust conclusions about the evolution of the cloud in that period. The same time would merely lead to two or three individually identified objects for 5-minute data output, which would be the minimum to draw any useful conclusions about the lifetimes or time evolution of the clouds.

## 4   Example B: tracking of deep convective clouds in model simulations and geostationary satellite data based on outgoing longwave radiation (OLR)

Satellite retrievals are an important tool in climate and weather research as they are an effective way of obtaining observation-based quantities over greater spatial scales in the atmosphere. Specifically, geostationary satellites offer continuous coverage in space and time for a specific region and can, therefore, be used for understanding the temporal evolution of atmospheric phenomena. Direct comparisons of model simulations with satellite retrievals for the same area and time period are an important means of assessing the models' capabilities to successfully represent atmospheric processes in the real world. Using a tracking framework for the analysis allows us to investigate the representation of clouds in the model in a way that takes the development of individual clouds within the population of clouds into account as opposed to relying on temporal and spatial statistics of the cloud field. Using the same tracking framework for both model and observation data allows for a more robust comparison between them.

Here, we use satellite data from the Geostationary Operational Environmental Satellite (GOES) system, specifically GOES-

13 (Hillger and Schmit, 2007), and WRF model simulation results from the ACPC deep convection case study (see Sect. 3). The satellite data were downloaded from the NOAA Comprehensive Large Array-data Stewardship System (CLASS) (NOAA, 2019) for the Continental United States (CONUS) area in NetCDF format. The NOAA Weather and Climate Toolkit (WCT) (National Climatic Data Center, NESDIS, NOAA, 2019) was used to convert pixel counts to radiances and brightness temperatures for the two channels used in the analysis here. The satellite data used in this example has an average horizontal spacing of about 4 km. The model simulation comprises the outermost grid of the nested WRF simulations used to drive the simulations in the inner domain that we used for the cloud tracking in the first example application (Sect. 3). This outer domain covers a much larger area, encompassing most of Texas and the surrounding states of the southern USA, as well as neighbouring areas of northeastern Mexico. The simulations were performed with a grid spacing of 4.5 km and used explicit convection without a cumulus parametrisation. The simulations have been performed for 24 hours from 12:00 UTC on the 19 June 2013 to 12:00 UTC on the 20 June 2013 and a square domain of 500 by 500 grid cells centred around Houston, Texas (van den Heever et al., 2017).

To be able to perform a meaningful comparison of the two datasets, it has to be ensured that the analysis covers the same region at a similar temporal and spatial resolution. The satellite data were restricted to the temporal and spatial extent as the model output data. Furthermore, the regular 15-minute interval available from the satellite data was used for both datasets, with a difference of up to a minute due to the scan time of the satellite data.

Top of the atmosphere outgoing longwave radiation (OLR) is used to track individual deep convective clouds in both model simulations and satellite retrievals. OLR is a standard model output for most high-resolution simulations and is often used as a diagnostic for simulated deep convection (Pearson et al., 2010; Russo et al., 2011). OLR retrievals also have the benefit that they do not depend on other aspects of a complicated radiative transfer model, which require, amongst other assumptions, that pixels are assigned as either cloud or cloud-free for the radiative retrieval of several optical (effective radius and optical depth) and thermal (cloud top temperature and height) cloud properties (McGarragh et al., 2018). For the satellite data, we use an empirical conversion derived in Singh et al. (2007) to convert the radiances $L$ from two channels in the GOES-13 measurements, the water vapour channel (WV, 5.8 to 7.30 µm) and a channel in the infrared window (WIN, 10.2 to 11.2 µm), to OLR.

$$\begin{aligned} OLR =& 11.44\,L_{\mathrm{WIN}} + 9.04\,L_{\mathrm{WV}} + \frac{9.11\,L_{\mathrm{WV}}}{L_{\mathrm{WIN}}} \\ & - \frac{86.36}{L_{\mathrm{WIN}}} - 0.14\,L_{\mathrm{WV}}^2 + 111.12\,. \end{aligned} \tag{4}$$

Singh et al. (2007) report an uncertainty from these conversions within 2.5 $\mathrm{W\,m^{-2}}$

The distribution of OLR for the model simulations and the satellite retrievals show a very similar shape (Fig. 6). The satellite-retrieved OLR features a larger number of pixels characterised by lower OLR values in the range between 100 and 250 $\mathrm{W\,m^{-2}}$ corresponding to deep cloud tops. The range covered and the peak position of OLR, corresponding to cloud-free and low cloud regions around 290 $\mathrm{W\,m^{-2}}$, agree well between the model simulation and the satellite retrieval.

We use these histograms to choose the threshold values for the feature detection and the segmentation steps in the tobac routine. The threshold for the outline of the convective clouds in the segmentation step (250 $\mathrm{W\,m^{-2}}$) reflects the lower tail of the peak of OLR in both the model simulations and the satellite retrievals. The additional thresholds used in the feature detection



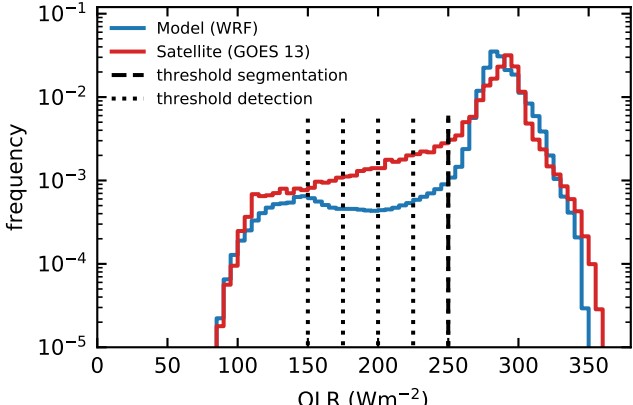

**Figure 6.** Probability density function of OLR for the model simulation and the satellite retrievals including the thresholds (vertical dashed and dotted lines) set for feature detection and segmentation.

algorithm (250, 225, 200, 175 and 150 W m$^{-2}$) are distributed over the range of OLR values in the part of the distribution representing the deeper clouds.

The individual steps of the tracking analysis for the model data are shown in Fig. 7, but the same steps are applied equally to the satellite-retrieved data. The outgoing longwave radiation field (Fig. 7a) is filtered with a Gaussian filter with a stan-
dard deviation of $\sigma = 4.5$ km, equivalent to the grid spacing of the model data (Fig. 7b). The feature identification following Sect. 2.2 is performed with the set of five OLR thresholds of 250, 225, 200, 175 and 150 W m$^{-2}$(Fig. 7c, d). The segmentation is performed using the watershedding technique (Sect. 2.3) with an OLR threshold of 250 W m$^{-2}$ to identify the area of the individual clouds leading to the cloud areas shown in 7e. The complete linked trajectories of all clouds present at the specific timestep, as illustrated in the other sub-figures, are shown in Fig. 7f with the cloud extent at the start (dashed) and end (solid)
of the lifetime of the cloud. A smaller subset of the data and analysis for this example including the tracking analysis and visualisation is available as a Jupyter notebook as part of the package source code. The tracked clouds for both the model simulation and the satellite retrieval are visualised for two different times in Fig. 8. Both the model simulations and the satellite retrieval show many individual convective clouds in a region north of the coastline, especially towards the east of the analysed domain around the Mississippi River Delta and further inland in Texas. In addition, larger connected regions of clouds occur
both towards the southern end of the analysed domain over the Gulf of Mexico and in the form of a large organised storm system entering the domain from the north-west. The propagation of this large system is not represented accurately in the model simulation, as it shows a lag of several hours and is smaller in magnitude than in the satellite retrievals. The lifetime distribution of the clouds identified and tracked from the model simulations and from the satellite retrievals show a similar distribution (see Fig. 9a). However, more clouds are identified in the satellite data than in the model data. When normalised
for total number, the lifetime distributions agree better between the two different data inputs (Fig. 9b). Most cloud objects are tracked for periods of up to an hour, but in both the model simulations and the satellite retrievals there are numerous cloud



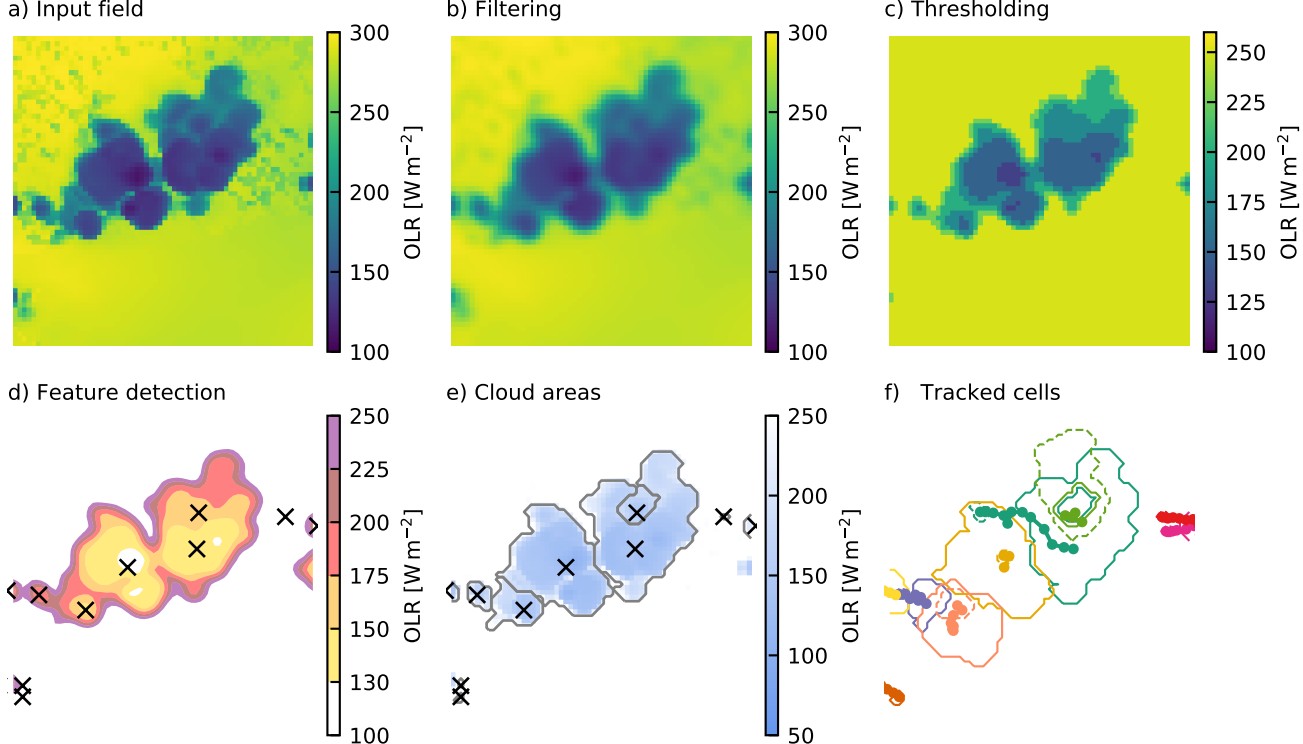

**Figure 7.** Schematic overview of the individual steps of the tracking algorithm for an example subset of the domain used in example B based on outgoing longwave radiation. The input data (a) are smoothed with a filter (b) before regions above or below a set of thresholds are determined (c) to identify the individual features (d). (e) shows the associated cloud areas determined in the segmentation step and (f) shows all individual clouds present at the time step over their entire life cycle, including outline the cloud area at the start (dashed) and at the end (solid) of the trajectory.

objects tracked for up to several hours. The distributions of the cloud areas (Fig. 9c, d) show that the total cloud area for both model and satellite data is made up of two types of identified objects, smaller tracked clouds with a radius of up to $100\,\mathrm{km}$ and large tracked features with a radius of a few hundred kilometres. Due to the larger number of tracked clouds, there is more total cloud area in the tracked clouds in the satellite data. The distribution of cloud sizes is relatively similar between the two datasets. The satellite data show more small clouds below $100\,\mathrm{km}$ equivalent radius. Furthermore, the size of the largest tracked objects is larger in the satellite data than in the model data, which corresponds to the large MCS propagating through the domain of interest (Fig. 8), and which is not represented properly in the model simulations with respect to both timing and total size.

An analysis of the cloud velocities and nearest-neighbour distances as described in Sect. 3.1 is presented in Fig. 10. The distribution of both the nearest-neighbour distances (Fig. 10a) and the cloud displacement velocities (Fig. 10b) agree well between the model simulations and the satellite retrieval. The peak of the nearest-neighbour distances appears around $20\,\mathrm{km}$. The prop-





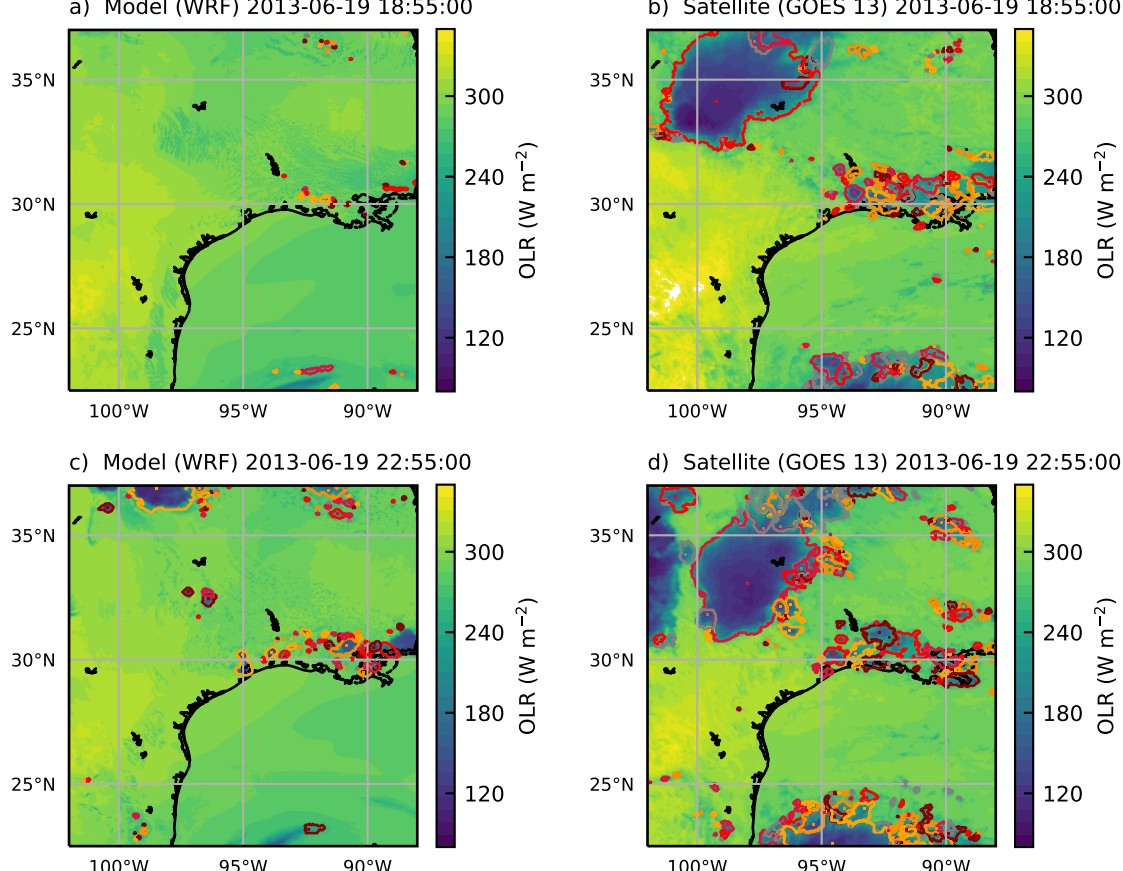

**Figure 8.** Identified and tracked objects at two specific points in time (19/06/2013 18:55 UTC and 22:55 UTC) based on outgoing long-wave radiation for the WRF simulations with 4.5 km grid spacing on the left (a, c) and the outgoing longwave radiation derived from the combination of two GOES-13 channels following Singh et al. (2007) on the right (b, d).





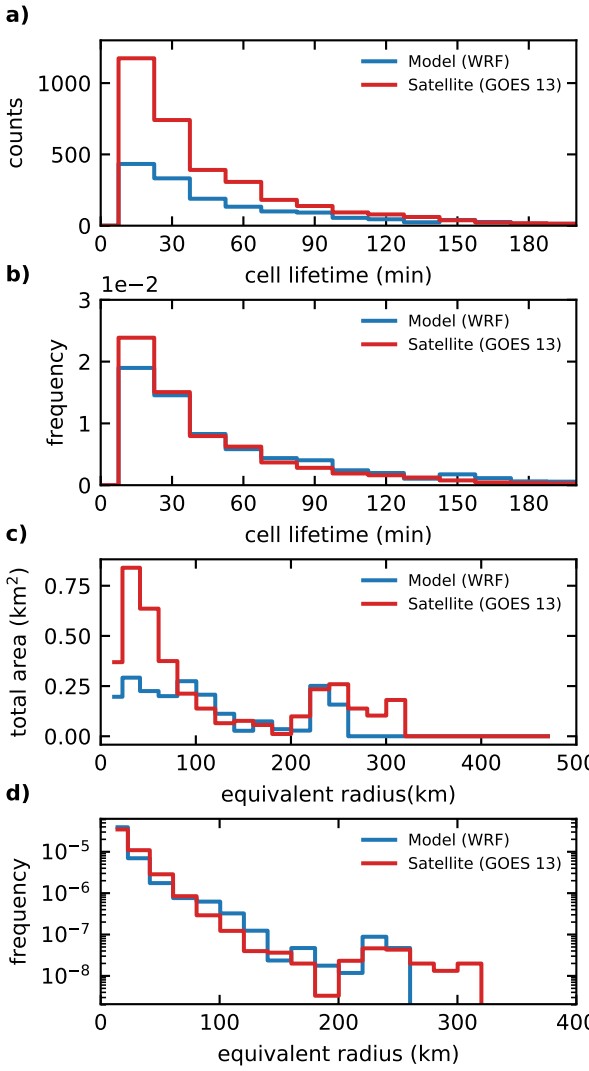

**Figure 9.** Distributions of cloud lifetimes obtained from the tracking of model data and satellite retrievals, shown as total counts (a) and frequency (b). The distribution of cloud areas is shown as the distribution of total area resulting from the sum in each area bin (d) and as a pdf of cloud area (c). Both these distributions are plotted against the equivalent radius of a circular cloud of the same area.





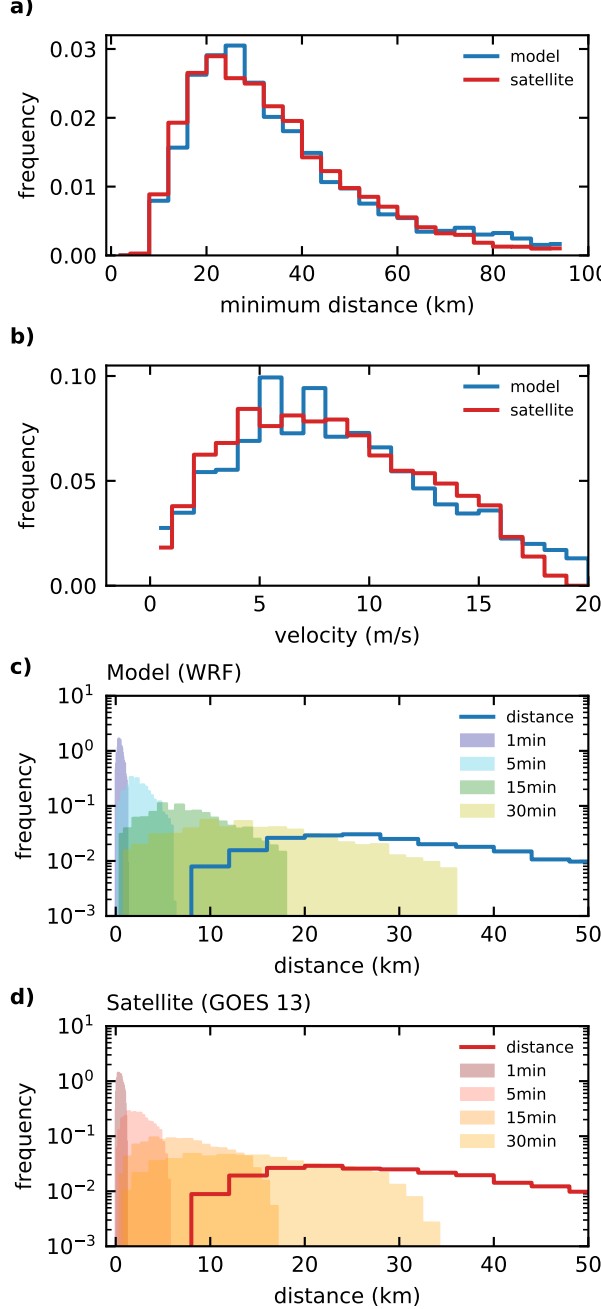

**Figure 10.** (a) Distributions of the distance to the next identified object for all identified objects and (b) velocities for tracked cloud objects at each timestep of the trajectories for both the model simulations and the satellite data. (d) The travel distance per input interval resulting from different time resolution of the input based on these velocities (b) is shown together with the distribution of the minimal distance to the nearest-neighbour in (a) for the model data in (c) and for the satellite data.





agation velocities peak at around $8\,\mathrm{m\,s^{-1}}$, with most of the velocities below $20\,\mathrm{m\,s^{-1}}$. A comparison of the nearest-neighbour distances and the displacements per input timestep that would result for different temporal resolution (1, 5, 15 and 30 minutes) shows that the 15-minute timestep used here already shows some overlap in the distributions. Longer timesteps of 30 minutes or more would probably lead to problems in the tracking, while shorter timesteps of a few minutes would be expected to im-

prove the tracking further. However, output at similarly high temporal frequencies is not always feasible or simply not available for a lot of data sources, e.g. for the GOES-13 geostationary satellite retrievals used in this study. The newest generation of geostationary satellite imagers such as the GOES-R series (GOES 16/17) that has replaced the GOES-13 satellite used here, as well as Himawari-8 (Bessho et al., 2016) and the future Meteosat Third Generation (MTG) satellites (Stuhlmann et al., 2005) all feature significantly higher temporal and spatial resolution.

The scattered convective cells of differing depths over the area of Houston that were the focus of the analysis in the first application example (Sect. 3) are not clearly resolved in these two datasets. The lower spatial resolution of the simulations and satellite retrieval (around $4\,\mathrm{km}$ compared to $500\,\mathrm{m}$ in the high-resolution simulations used in Sect. 3) limit the spatial scale of cloud features that can be resolved to more than a few tens of kilometres in radius. The use of outgoing longwave radiation as a variable for feature identification does not include as much information as the three-dimensional model output fields used in

Sect. 3, however, it provides complementary information to compare model simulations with satellite retrievals.

## 5 Conclusions

We have presented tobac, a new framework for object-based analysis and tracking individual convective clouds in different types of input data. The workflow of the software package consists of the detection of suitable features, segmentation of the

areas or volumes representative of an individual cloud object and subsequent linking of objects at individual time steps into trajectories. All individual steps are implemented in a modular way, thereby allowing for the implementation of different algorithms for each of the steps, should the need arise.

We have developed a feature detection algorithm based on identifying regions above/below a defined sequence of thresholds in two-dimensional input fields. Cloud volumes or cloud areas are associated based on a watershedding technique featuring a

single specific threshold value on two- or three-dimensional input fields.

We have shown how we can leverage another open-source Python package trackpy, initially developed for application in microscopy, in the tobac framework to link up cloud objects at individual timesteps into consistent cloud trajectories. These cloud trajectories allow for an analysis of cloud lifetimes and the time evolution of cloud properties and physical processes in the clouds over the lifetime of the cloud. The analysis routines provided as part of the package can be applied to derive cloud

properties and statistics for individual clouds over their life-cycle as well as for the entire population of clouds in the analysed cloud field. The built-in visualisation routines allow for a convenient way to assess the performance of the analysis and evaluate the choice of parameters for the different steps of the analysis framework. The automatically created animated visualisations of individual tracked cells can guide users in the development of further detailed analyses based on the analysis tools provided



in the framework.

The implementation of the tracking framework in Python enables the use of extensive and actively developed open source libraries for scientific computing. We have shown that this provides numerous advantages, e.g. for memory management, data structures and visualisation. The rapid development of the underlying libraries means that tobac can profit from future advances

without any further development of tobac and any requirements on the side of the user.

The modular structure of the framework allows for the inclusion of other existing or newly developed methods for the individual steps of feature detection, object segmentation and tracking into the software package in the future. These capabilities enable the use of different tracking algorithms in parallel for evaluation and comparisons as well as tracking based on different types of input data in a single analysis framework.

We have presented two application examples of the use of tobac for the study of deep convective clouds. In the first application (example A), we have tracked scattered deep convective cells based on a combination of the vertical velocity and total condensate mixing ratio fields from CRM simulations with WRF over the area around Houston, Texas. The simulations were performed with a grid spacing of $500\,\mathrm{m}$, and thus represent a typical application of a CRM. The tracking framework is currently being applied to other CRMs for the same case study as part of the ACPC deep convection case study (van den Heever et al.,

2017) to investigate the response of deep convective clouds in models to changes in aerosols. We have performed the tracking for different output frequencies to evaluate the dependency of the tracking performance on the time resolution of the input data. The output resolutions of 1 minute and 5 minutes lead to comparable tracking results for scattered convective cells. This result can be confirmed using an analysis of typical displacement velocities of the clouds and nearest-neighbour distances between the individual identified cloud objects.

In a second application (example B), we have presented a simultaneous tracking of deep convective cloud features and larger convective systems based on outgoing longwave radiation output from model simulations with convection-permitting grid spacing ($4.5\,\mathrm{km}$) and outgoing longwave radiation derived from geostationary satellite retrievals (GOES-13) in the same region. The 15-minute time resolution available from the satellite retrieval is shown to be sufficient for successful tracking performance. The analysis also demonstrated that the model simulations and the satellite retrieval feature clouds with a similar lifetime dis-

tribution. The distribution of cloud areas in model and satellite data shows a similar combination of smaller convective cells and larger systems. The main differences occur for the largest tracked systems, which are stronger in the satellite retrievals. This can be explained by the limited representation of the propagation of two large organised storms within the model domain. This would have been more challenging to assess from a bulk analysis of the domain-wide averaged properties.

The newest generation of geostationary satellites, such as Himawari-8 or GOES-16/17, provide significantly higher spatial and

temporal resolution (Bessho et al., 2016; Schmit et al., 2016). These advances will strongly improve the applicability of this type of satellite data for use in object-based tracking and analyses with tobac, and also allows for a wider range of applications, e.g. by capturing smaller scattered cells such as the ones investigated in Sect. 3.

The ability of tobac to be used for both models and observations as shown in these examples helps to more directly compare models with observations, and therefore, better understand the differences between the two types of data.

Although we have focused on tracking and analysing deep convection here, there are numerous other applications that tobac



can be used for without much additional work. There are a large number of existing data products, such as high-resolution radar data, e.g. from NEXRAD over the United States or similar networks in several other regions of the world (Reed et al., 2017), that would be most suited for the use with tobac. Furthermore, the application of tobac is not strictly limited to the analysis of clouds, and it can also be applied to study other features of the Earth system that can be identified as well-defined

time evolving regions, such as distinct aerosol plumes in the atmosphere or plankton in the surface layer of the ocean.

We are currently working on implementing additional algorithms for the modular steps of the framework, e.g. based on the analyses developed in Senf et al. (2018). Additionally, we are implementing a more flexible representation of the links between cloud objects at specific points in time, which will allow for a proper treatment of more complex splitting and merging of cells. We invite the community to contribute to the future development of tobac both through the implementation of existing

algorithms into the common framework and by using the framework as a basis for new developments.

*Code and data availability.* The tobac source code publicly available in a GitHub repository distributed under a BSD 3-Clause license at https://github.com/climate-processes/tobac (Heikenfeld, 2019c, d). The version described here is available as a release (Heikenfeld, 2019a). The linking step makes use of trackpy (Allan et al., 2016), which is available on GitHub (https://github.com/soft-matter/trackpy). We use several standard Python packages for scientific computing and image processing that are all available through package managers such as pip

or conda. The GOES-13 satellite imager data has been downloaded from the NOAA Comprehensive Large Array-data Stewardship System (CLASS) (NOAA, 2019) and processed with the NOAA Weather and Climate Toolkit (WCT) (National Climatic Data Center, NESDIS, NOAA, 2019).

Jupyter notebooks containing the tracking analysis and visualisations of the tracking results for a smaller subsample of the data used in the two example applications are provided as part of the tobac source code. The data used in these notebooks is downloaded automatically and

is avaliable as Heikenfeld (2019b) and on GitHub at https://github.com/climate-processes/tobac_example_data.

*Author contributions.* M. Heikenfeld developed tobac with contributions by P.J. Marinescu, D.Watson.-Parris and F. Senf.. M. Christensen processed the geostationary satellite data and contributed to the analysis of the tracking based on outgoing longwave radiation. M. Heikenfeld performed the data analysis and wrote the manuscript with contributions and approval of the final version by P.J. Marinescu., P. Stier., S.C. van den Heever, M. Christensen., D. Watson.-Parris and F. Senf.

*Acknowledgements.* M. Heikenfeld acknowledges funding from the NERC Oxford DTP in Environmental Research (NE/L002612/1). The research leading to these results has received funding from the European Union's Seventh Framework Programme (FP7/2007-2013) project BACCHUS under grant agreement n° 603445 and the European Research Council under the European Union's Horizon 2020 research and innovation program with Grant Agreement n° 724602 (RECAP). S.C. van den Heever and P.J. Marinescu acknowledge funding from the NASA CAMPEx project under grant 80NSSC18K0149, and P.J. Marinescu was supported by a National Science Foundation Graduate

Research Fellowship Grant DGE-1321845. F. Senf acknowledges funding within the High Definition Clouds and Precipitation for advancing Climate Prediction (HD(CP)$^2$) project funded by the BMBF (German Ministry for Education and Research) under grant 01LK1507C.



The research has been performed in the framework of the IGBP/WCRP initiative ACPC (Aerosols Clouds Precipitation and Climate) with a focus on developing analysis techniques for the deep convection case study over Houston, Texas. Computations and data processing have been performed on the ARCHER and JASMIN computing facilities.





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
