# Peer review of "tobac 1.2: towards a flexible framework for tracking and analysis of clouds in diverse datasets"

_Geoscientific Model Development, 2019_

## Referee Comment (RC1) · Anonymous Referee #1 · 25 Jun 2019

This paper present a tracking code for clouds and other atmospheric objects, using modern programming techniques and languages. The topic is certainly not new, but the fact that the code is developed to be flexible and openly available warrants a code description paper. I find the paper clearly written, appreciate the examples, and recommend publication after the following minor questions are answered:

1) Was it a conscious decision to not tack on ". . . and COnvection" to the acronym?

2) What is the performance of this algorithm? For example, for each of the test cases, what was the maximum memory footprint, runtime of the script, etc

3) It makes sense to separate the several stages of the workflow from a coding point of

view; is it also possible to save data after the trajectory linking, and if so, in what format and what kind of design?

4) P6: Do I understand correctly that the feature detection only works in 2D? Dawe and Austin (and others) did 3D feature detection, which ends up being more precise, especially also in the tracking part.

5) P9, l20: A circular range parameter v_max suggests that the trajectory linking is performed isotropically. Is it possible to assign/automatically retrieve an advection velocity as well? Heiblum et al did this very effectively.

---

## Referee Comment (RC2) · Anonymous Referee #2 · 4 Jul 2019

**1   Reviewer's summary of the manuscript**

In "tobac v1.0: towards a flexible framework for tracking and analysis of clouds in diverse datasets," Heikenfeld et al. present a new, open source software framework for detecting and tracking clouds in datasets ranging from satellite observations to model output. They describe the details and justification for their Python-based implmentation, which extensively leverages existing libraries that offer a number of benefits (e.g., lazy data loading from dask), and they describe in detail the various algorithmic steps of tobac. They further provide two illustrating examples of how tobac might be used: tracking of deep convective clouds in 3D CRM output and tracking/comparison of clouds in

2D satellite data and 2D model output. The authors also discuss some limitations of tobac–especially restrictions on the appropriate temporal resolution of the input data, and they describe how the modular design of tobac will facilitate the authors (and/or future users) to easily improve/upgrade various algorithmic components.

**2  Summary of Review**

Overall, Heikenfeld et al. present a well-written and thorough description of a new, open-source framework that other geoscientists will likely find useful. The paper is well within the scope of GMD. The author's algorithmic choices are both well-explained and justified, and the methods by which the authors evaluate tobac are sound: particularly with respect to evaluation of the impact of temporal resolution.

I ordinarily have many paragraphs of feedback when reviewing papers, but in this case, the authors have put forth a solid manuscript; I do not have much feedback to offer, beyond a suggestions for clarifying in a few places. Based on this assessment, I recommend that the manuscript be accepted for publication in GMD pending some minor revisions. The authors should be commended on assembling an excellent manuscript and describing a useful open-source code.

**3  General Comments**

**3.1  Some additional citations and discussion**

In section 1, the authors give a reasonably thorough overview of other examples of cloud tracking in the literature. There are a few additional references that I would suggest the authors discuss:

- Wilcox, Eric M. "Spatial and Temporal Scales of Precipitating Tropical Cloud Systems in Satellite Imagery and the NCAR CCM3." Journal of Climate 16, no. 22 (November 1, 2003): 3545–59. https://doi.org/10.1175/1520-0442(2003)016<3545:SATSOP>2.0.CO;2.

Note that the following papers focus on identifying clouds in time-slices, and not necessarily tracking them in time. But the concept is close enough–the papers are essentially steps 1–3 in the tobac workflow (Figure 1)–that these papers warrant discussion.

- Wilcox, Eric M, and V Ramanathan. "Scale Dependence of the Thermodynamic Forcing of Tropical Monsoon Clouds: Results from TRMM Observations." Journal of Climate 14, no. 7 (April 1, 2001): 1511–24. https://doi.org/10.1175/1520-0442(2001)014<1511:SDOTTF>2.0.CO;2.

- Wood, Robert, and Paul R. Field. "The Distribution of Cloud Horizontal Sizes." Journal of Climate 24, no. 18 (2011): 4800–4816. https://doi.org/10.1175/2011JCLI4056.1.

- O'Brien, Travis A, Fuyu Li, William D Collins, Sara A Rauscher, Todd D Ringler, Mark Taylor, Samson M Hagos, and L Ruby Leung. "Observed Scaling in Clouds and Precipitation and Scale Incognizance in Regional to Global Atmospheric Models." Journal of Climate 26, no. 23 (December 2013): 9313–33. https://doi.org/10.1175/JCLI-D-13-00005.1.

- Igel, Matthew R., Aryeh J. Drager, and Susan C. van den Heever. "A CloudSat Cloud Object Partitioning Technique and Assessment and Integration of Deep Convective Anvil Sensitivities to Sea Surface Temperature." Journal of Geophysical Research: Atmospheres 119, no. 17 (2014): 10515–35. https://doi.org/10.1002/2014JD021717.

- Guillaume, A., B. H. Kahn, Q. Yue, E. J. Fetzer, S. Wong, G. J. Manipon, H. Hua, and B. D. Wilson. "Horizontal and Vertical Scaling of Cloud Geometry Inferred from CloudSat Data." Journal of the Atmospheric Sciences 75, no. 7 (July 2018): 2187–97. https://doi.org/10.1175/JAS-D-17-0111.1.

As a side note, I am a bit surprised that one of these references wasn't already included, as it was written by one of the co-author's former Ph.D. students.

In addition to these additional references, it might be useful to add a paragraph or two that describes what–in terms of science–was has been learned by using cloud tracking. For readers unfamiliar with cloud tracking, this might help justify the scientific motivation for a flexible and open-source software package for cloud tracking.

**3.2 Feature ID vs Segmentation**

Perhaps I'm being a bit dense about this, but I re-read both sections 2.2 and 2.3 several times and could not determine the functional difference between feature identification and segmentation.

Based on what I'm reading, it sounds like the only output of the feature identification step is the set of feature positions (weighted mean centers), from which the segmentation starts. Is this understanding correct? If so, this point should be emphasized, and if not, the text would benefit from a revision to make the distinction between feature identification and segmentation more clear.

In later parts of the manuscript, it seems that one functional difference might be that feature identification and segmentation might use different variables: e.g., max vertical velocity vs cloud condensate mixing ratio. Would the authors get the same result if they segmented the condensate field, based on any values above the segmentation threshold, and then filtered out objects with max vertical velocity below a certain threshold?

I'm not suggesting that this should be done, but rather I am illustrating that it might be useful to discuss the feature-ID/segemntation approach versus other plausible approaches; such a comparison might help readers grasp what appears (to me) to be a subtle distinction.

**3.3 Style of package/software names**

The manuscript contains a lot of references to software libraries, e.g.,: tobac, pandas, xarray, scipy, etc.. It is good, and useful, that the authors do this, but it might be useful to use a different text style for these software package names in order to visually distingush package names from English words. This is important to make the manuscript accessible to readers who might not be familiar with the Python ecosystem of packages, and it is especially important for software packages that could easily be confused for English words. For example, on pg 10, line 8, the sentence "The trajectories are recorded in a pandas DataFrames" might trip-up a non-Python-initiated reader (or at least amuse them). If the authors used LaTeX to compose the manuscript, and if GMD style rules allow it, I would strongly suggest that the authors use the `\verb:package-name:` macro.

**3.4 Nice code**

This is a compliment rather than constructive feedback: I appreciate the pervasive use of comments in the code and the clear and consistent documentation of functions. It was easy to skim through the code and get a general understanding of how the code is functioning.

**4 Minor issues**

p 6, line 25-26: The use of the term 'erosion' here is a bit unclear: the use of this word, in this sense, is not common in the geophysical sciences. The term 'erosion' should be explained here.

p 7, lines 6-16: this multithreshold approach is interesting. I don't think I've seen this in the literature before - do the authors have a reference for this, or is this an innovation of this study? It should be stated either way.
Also, it might be useful to state whether the framework is flexible enough to permit a single threshold value, which might be appropriate in cases where a person wants to track any contiguous feature with non-zero cloud condensate.

p10 lines 2-5: "The trajectories.." <– it isn't clear to me what this sentence means. It might need to be rephrased.

p 10, lines 12-13: "Instead, the algorithm. . . " it might be useful to produce an illustration of this. I'm finding that I'm unable to visualize how cloud trajectories from tobac would appear when there are cloud splits/mergers

---

## Author Comment (AC1) · 14 Sep 2019

We would like to thank the reviewer for their comments and suggestions. The review pointed out important aspects that we have clarified or addressed by minor changes in the manuscript. In the following, we respond to the reviewer's comments in black, with our answers to the comments in blue and the adapted text from the revised manuscript in green. We have attached the revised version of the manuscript with tracked changes to the general authors' response. In the general authors' response (AR), we have added a few additional comments regarding the revised manuscript and points raised by both reviewers.

**This paper present a tracking code for clouds and other atmospheric objects, using modern programming techniques and languages. The topic is certainly not new, but the fact that the code is developed to be flexible and openly available warrants a code description paper. I find the paper clearly written, appreciate the examples, and recommend publication after the following minor questions are answered:**

**1) Was it a conscious decision to not tack on ". . . and COnvection" to the acronym?**

We actually played with the idea of that rather obvious acronym. But apart from avoiding possible negative associations with the health impacts of the associated substance, we decided to stick to a unique name for the package to make it easier in terms of package managers, documentation, searchability, etc. and thus opted for the acronym *tobac*.

**2) What is the performance of this algorithm? For example, for each of the test cases, what was the maximum memory footprint, runtime of the script, etc**.

We want to thank the reviewer specifically for this comment on a very important aspect of the software package, that we had not addressed quantitatively in the initial version of the manuscript. In fact, creating the detailed performance analyses for the example cases and some puzzling results lead us to more detailed profiling that, especially in the code for the feature detection, allowed to reduce the processing time by more than an order of magnitude. These changes were related to time-consuming operations related to creating specific data structures or writing values into these data structures and do not change the analysis results at all. As part of the testing and the improvements, we have modularised the code for the feature detection and segmentation into several sub-functions, which makes the code more accessible and also more suitable for future adaptions and additions.

We have added information about the typical performance for the examples in the manuscript text, taking both calculations on a typical laptop and on a dedicated data analysis facility (JASMIN) into account:

"The data processing has been performed on the JASMIN data analysis facility (CEDA, 2019). The script including feature detection, segmentation, trajectory linking and saving of the analysis output for the 1-min data output had a processing time of around 17 minutes with a maximum memory footprint of 3.1 GB using a maximum of 3 processes and 27

threads. The segmentation step has been broken up into chunks of 10 minutes each to limit the total memory consumption of the analysis. The processing time is almost entirely taken up by the segmentation step using time-resolved 3-dimensional data of the total condensate. It is thus strongly affected by the time required to access the data on the disk and thus highly dependent on both the infrastructure and the structure of the data file, i.e. the data compression in the input files."
(page 14, line 10)

"The data processing was performed on the JASMIN data analysis facility (CEDA, 2019). The total processing time of the script including feature detection, segmentation trajectory linking and saving the output data was around 1 minute with a maximum memory footprint of around 500 MB for the model data and 2.5 minutes with a memory footprint of around 400 MB for the satellite data, each using a maximum of 3 processes and 27 threads."
(page 18, line 28)

**3) It makes sense to separate the several stages of the workflow from a coding point of view; is it also possible to save data after the trajectory linking, and if so, in what format and what kind of design?**

The individual stages of the workflow are separated as discussed in the "software description" section. Intermediate results can be saved after the initial steps in the format of *pandas* DataFrames for features and cloud tracks or as *iris* cubes in the case of the cloud masks resulting from the segmentation step. We have added that explicit information in the "software description" section of the manuscript:

"The intermediate results of each individual analysis step can be conveniently saved and examined in the form of pandas DataFrames or iris cubes."
(page 6, line 23)

**4) P6: Do I understand correctly that the feature detection only works in 2D? Dawe and Austin (and others) did 3D feature detection, which ends up being more precise, especially also in the tracking part.**

The feature detection we have currently implemented only works in 2D. This was a reasonable choice, especially with regard to the deep convective clouds we mainly focused on here. As you mention, there will be situations such as the analyses of shallow convection you referred to, where 3D identification and 3D tracking will be beneficial. We have concrete plans to include 3D feature detection and tracking in the next major version of tobac.

**5) P9, l20: A circular range parameter v_max suggests that the trajectory linking is performed isotropically. Is it possible to assign/automatically retrieve an advection velocity as well? Heiblum et al did this very effectively.**

The advection velocity of the trajectories in previous time steps is actually used as part of the tracking using `trackpy`. Thus, v_max only describes the size of a circle around the predicted position based on the advection in the previous steps, i.e. the allowed deviation of from the predicted position using this information from the previous time steps. We have adapted the manuscript in this section to make this clearer and added a schematic depiction of a specific linking step as Figure 3.

"The linking determines which of the features detected in a specific time step (see Sect. 2.2) is identical to an existing feature in the previous time step and is illustrated in Fig. 3. For each existing feature, the movement within a time step is predicted based on the velocities in a number of previous time steps. The algorithm then breaks the search process down to candidate features by restricting the search to a circular search region centred around the predicted position of the feature in the next time step. For newly initialised trajectories, where no velocity from previous time steps is available, the algorithm resorts to the average velocity of the nearest tracked objects. The parameter v max restricts how much the future position of a feature is allowed to deviate from a linear extrapolation of the trajectory over time. It thus has the units of a velocity and describes the dependency of the circular search range d on the output time step $\Delta t$ in the data used for the tracking $d = v \ max \ \Delta t$."
(page 9, line 31)

[Figure]

**Figure 3**. Schematic illustration of the trajectory linking with the predicted motion of the feature based on previous time steps and a search
range around the predicted position.
(page 10)

---

## Author Comment (AC2) · 14 Sep 2019

**Authors Comments:**

We would like to thank the reviewer for their comments and suggestions as well as the motivating positive feedback on the manuscript and the software. The review pointed out important aspects that required additional clarity or information that we have clarified or addressed by changes in the manuscript. In the following, we respond to the reviewer's comments in black, with our answers to the comments in blue and the adapted text from the revised manuscript in green. We have attached the revised version of the manuscript with tracked changes to the general authors' response. In the general authors' response (AR), we have added a few additional comments regarding the revised manuscript and points raised by both reviewers.

**Anonymous Referee #2**

**1 Reviewer's summary of the manuscript**
In "tobac v1.0: towards a flexible framework for tracking and analysis of clouds in diverse datasets," Heikenfeld et al. present a new, open source software framework for detecting and tracking clouds in datasets ranging from satellite observations to model output. They describe the details and justification for their Python-based implmentation, which extensively leverages existing libraries that offer a number of benefits (e.g., lazy data loading from dask), and they describe in detail the various algorithmic steps of tobac. They further provide two illustrating examples of how tobac might be used: tracking of deep convective clouds in 3D CRM output and tracking/comparison of clouds in 2D satellite data and 2D model output. The authors also discuss some limitations of tobac–especially restrictions on the appropriate temporal resolution of the input data, and they describe how the modular design of tobac will facilitate the authors (and/or future users) to easily improve/upgrade various algorithmic components.

**2 Summary of Review**
Overall, Heikenfeld et al. present a well-written and thorough description of a new, open-source framework that other geoscientists will likely find useful. The paper is well within the scope of GMD. The author's algorithmic choices are both well-explained and justified, and the methods by which the authors evaluate tobac are sound: particularly with respect to evaluation of the impact of temporal resolution. I ordinarily have many paragraphs of feedback when reviewing papers, but in this case, the authors have put forth a solid manuscript; I do not have much feedback to offer, beyond a suggestions for clarifying in a few places. Based on this assessment, I recommend that the manuscript be accepted for publication in GMD pending some minor revisions. The authors should be commended on assembling an excellent manuscript and describing a useful open-source code.

**3 General Comments**

**3.1 Some additional citations and discussion**

In section 1, the authors give a reasonably thorough overview of other examples of cloud tracking in the literature. There are a few additional references that I would suggest the authors discuss:

- **Wilcox, Eric M. "Spatial and Temporal Scales of Precipitating Tropical Cloud Systems in Satellite Imagery and the NCAR CCM3." Journal of Climate 16, no. 22 (November 1, 2003): 3545–59. https://doi.org/10.1175/1520-0442(2003)016<3545:SATSOP>2.0.CO;2.**

Note that the following papers focus on identifying clouds in time-slices, and not necessarily tracking them in time. But the concept is close enough–the papers are essentially steps 1–3 in the tobac workflow (Figure 1)–that these papers warrant discussion.

- **Wilcox, Eric M, and V Ramanathan. "Scale Dependence of the Thermodynamic Forcing of Tropical Monsoon Clouds: Results from TRMM Observations." Journal of Climate 14, no. 7 (April 1, 2001): 1511–24. https://doi.org/10.1175/1520-0442(2001)014<1511:SDOTTF> 2.0.CO;2.**
- **Wood, Robert, and Paul R. Field. "The Distribution of Cloud Horizontal Sizes."Journal of Climate 24, no.18 (2011): 4800–4816. https://doi.org/10.1175/2011JCLI4056.1.**
- **O'Brien, Travis A, Fuyu Li, William D Collins, Sara A Rauscher, Todd D Ringler,Mark Taylor, Samson M Hagos, and L Ruby Leung. "Observed Scaling in Clouds and Precipitation and Scale Incognizance in Regional to Global Atmospheric Models." Journal of Climate 26, no. 23 (December 2013): 9313–33.https://doi.org/10.1175/JCLI-D-13-00005.1.**
- **Igel, Matthew R., Aryeh J. Drager, and Susan C. van den Heever. "A Cloud-Sat Cloud Object Partitioning Technique and Assessment and Integration of Deep Convective Anvil Sensitivities to Sea Surface Temperature." Journal of Geophysical Research: Atmospheres 119, no. 17 (2014): 10515–35. https://doi.org/10.1002/2014JD021717.**
- **Guillaume, A., B. H. Kahn, Q. Yue, E. J. Fetzer, S. Wong, G. J. Manipon, H. Hua, and B. D. Wilson. "Horizontal and Vertical Scaling of Cloud Geometry Inferred from CloudSat Data." Journal of the Atmospheric Sciences 75, no. 7 (July 2018):2187–97. https://doi.org/10.1175/JAS-D-17-0111.1.**

As a side note, I am a bit surprised that one of these references wasn't already included, as it was written by one of the co-author's former Ph.D. students.

We want to thank the reviewer for pointing us to these two aspects of the existing literature that had not been covered in the introduction section in the initial version of the manuscript. We have added a discussion of the investigation of cloud size statistics and the use of active sensors:

"The identification of individual cloud objects in satellite data is one aspect of cloud tracking and has been used to investigate the spatial scaling of clouds on a global scale (Wilcox and Ramanathan, 2001; Wood and Field, 2011), including studies on the representation of these distributions in global atmospheric models (Wilcox, 2003; O'Brien et al., 2013). The use of satellite data from active sensors (Nesbitt et al., 2000; Bacmeister and Stephens, 2011; Riley et al., 2011; Igel et al., 2014; Guillaume et al., 2018) allows to

include information about the vertical extent of identified cloud objects, which provides improved classification of cloud types and understanding of important physical processes." (page 2, line 30)

This includes the following two additional references in addition to the ones suggested by the reviewer:

Nesbitt, Stephen W., Edward J. Zipser, and Daniel J. Cecil. 2000. 'A Census of Precipitation Features in the Tropics Using TRMM: Radar, Ice Scattering, and Lightning Observations'. Journal of Climate 13 (23): 4087–4106. https://doi.org/10.1175/1520-0442(2000)013<4087:ACOPFI>2.0.CO;2.

Bacmeister, J. T., and G. L. Stephens. 2011. 'Spatial Statistics of Likely Convective Clouds in CloudSat Data'. Journal of Geophysical Research: Atmospheres 116 (D4). https://doi.org/10.1029/2010JD014444.

**In addition to these additional references, it might be useful to add a paragraph or two that describes what–in terms of science–was has been learned by using cloud tracking. For readers unfamiliar with cloud tracking, this might help justify the scientific motivation for a flexible and open-source software package for cloud tracking.**

We have amended the paragraph to add some additional summary of the scientific understanding gained from the use of cloud tracking:

"This overview clearly shows the wide range of extensive efforts that went into the development of elaborate software and analysis tools to track clouds in different types of datasets. The application of cloud identification and tracking and related techniques has substantially increased our understanding of cloud size distributions, the time evolution of different types of clouds, and the underlying physical processes governing cloud formation, development, and propagation. However, the overview also highlights the problem of limited compatibility between the different existing approaches and implementations, especially regarding the intended use of tracking clouds based on different data sources using the same algorithms and analysis tools."
(page 3, line 27)

**3.2 Feature ID vs Segmentation**

**Perhaps I'm being a bit dense about this, but I re-read both sections 2.2 and 2.3 several times and could not determine the functional difference between feature identification and segmentation. Based on what I'm reading, it sounds like the only output of the feature identification step is the set of feature positions (weighted mean centers), from which the segmentation starts. Is this understanding correct? If so, this point should be emphasized, and if not, the text would benefit from a revision to make the distinction between feature identification and segmentation more clear. In later parts of the manuscript, it seems that one functional difference might be that feature identification and segmentation might use different variables: e.g., max vertical velocity vs cloud condensate mixing ratio.**

The reviewer has understood the procedure the right way regarding feature detection and segmentation. In the current implementation of tobac, the positions of the detected features are the only information passed on to the next steps segmentation and trajectory linking.
Separating the two steps allows for the use of different datasets as shown in Sect. 3. In addition, the separation of the two steps makes sense in the case of three-dimensional input data, as the tracking is currently restricted to two-dimensional input.
We have worked over the description to make this clearer in the revised manuscript, e.g. at the end of the description of the segmentation step:

"Similarities between the feature detection and segmentation steps mean that these steps could be combined in some implementations in future versions of tobac, e.g. for applications based on a single input dataset as the one used in Sect. 4. However, treating the two analysis steps separately allows for the combination of different datasets as shown in Sect. 3."
(page 9, line 21)

**Would the authors get the same result if they segmented the condensate field, based on any values above the segmentation threshold, and then filtered out objects with max vertical velocity below a certain threshold?**
**I'm not suggesting that this should be done, but rather I am illustrating that it might be useful to discuss the feature-ID/segmentation approach versus other plausible approaches; such a comparison might help readers grasp what appears (to me) to be a subtle distinction.**

We have not tried this specific order of operations suggested by the reviewer. However, tests of using just the condensate field for detection of individual clouds were not really successful in the case of tracking deep convective clouds. This is related to the fact that the condensate field is substantially less confined compared to the comparatively well defined narrow updraft regions and also much more interconnected between neighbouring cloud features (e.g., via anvils or cold pool boundaries).
.

**3.3 Style of package/software names**

**The manuscript contains a lot of references to software libraries, e.g.,: tobac, pandas, xarray, scipy, etc.. It is good, and useful, that the authors do this, but it might be useful to use a different text style for these software package names in order to visually distingush package names from English words. This is important to make the manuscript accessible to readers who might not be familiar with the Python ecosystem of packages, and it is especially important for software packages that could easily be confused for English words. For example, on pg 10, line 8, the sentence "The trajectories are recorded in a pandas DataFrames" might trip-up a non-Python-initiated reader (or at least amuse them).**

**If the authors used LaTeX to compose the manuscript, and if GMD style rules allow it, I would strongly suggest that the authors use the \verb:package-name: macro.**

We have opted for typewriter-style font for the names of Python packages such as `Iris`, `xarray` or `dask` to make these manuscript clearer and easier to read for readers not familiar with the specific Python packages names.

**3.4 Nice code**

**This is a compliment rather than constructive feedback: I appreciate the pervasive use of comments in the code and the clear and consistent documentation of functions. It was easy to skim through the code and get a general understanding of how the code is functioning.**

**Minor issues**

**p 6, line 25-26: The use of the term 'erosion' here is a bit unclear: the use of this word, in this sense, is not common in the geophysical sciences. The term 'erosion' should be explained here.**

We have extended the description for the use of the erosion techniques from image processing in more detail and explained how they are used in the context of the tobac feature detection:

"The detection of regions above a specific threshold can lead to large interconnected regions combining several features linked by narrow ridges. To prevent this and identify these interconnected features separately, the tobac feature detection allows for the use of "erosion" techniques based on the implementation in `skimage.morphology.binary_erosion`. These techniques shrink the identified regions from the edges by a specific length or number of pixels, thus removing the connecting ridges between interconnected features. This has been shown to lead to more robust detection of individual features, as described in detail in Senf et al. (2018)."
(page 6, line 31)

In a similar way we have also added a more detailed definition of 'watershedding', which clould lead to misunderstandings with readers unfamiliar with image processing techniques:

"Watershedding segmentation treats the input field as a topographic map and separates the input into individual regions similar to individual watersheds or catchment basins along a dividing ridge in a geological context (Meyer, 1994). These techniques are widely used in several existing cloud tracking and analysis algorithms described in Sect. 1, such as Heiblum et al. (2016a), Fiolleau and Roca (2013) and Senf et al. (2018)."
(page 9, line 7)

**p 7, lines 6-16: this multithreshold approach is interesting. I don't think I've seen this in the literature before - do the authors have a reference for this, or is this an innovation of this study? It should be stated either way. Also, it might be useful to state whether the framework is flexible enough to permit a single threshold value, which might be appropriate in cases where a person wants to track any contiguous feature with non-zero cloud condensate.**

We have not based our approach on any existing literature in the field of cloud tracking. The multi-threshold methodology was developed as an extension of working with a single threshold, which is still perfectly possible, so we have stated this explicitly in the revised manuscript. There are however other approaches using an iterative set of threshold values for cloud detection in a different way along with examples from another field of science with similar challenges, such as astronomy. We have thus added references to these other types of multi-threshold feature detection methods:

"While using multiple thresholds is usually beneficial, feature detection using a single threshold value is possible in tobac by only supplying a single threshold value and can be appropriate in certain applications. An iterative set of threshold values was used in other recent approaches to cloud detection (Liang et al., 2017; Fu et al., 2019), however with a specific focus on the application to certain types of satellite images. Multiple-threshold methods are also applied in other fields of science facing similar challenges in feature detection such as astronomy (Zheng et al., 2015)."
(page 7, line 27)

Liang, Kuai, Hanqing Shi, Pinglv Yang, and Xiaoran Zhao. 2017. 'An Integrated Convective Cloud Detection Method Using FY-2 VISSR Data'. Atmosphere 8 (2): 42. https://doi.org/10.3390/atmos8020042.

Fu, Hualian, Yuan Shen, Jun Liu, Guangjun He, Jinsong Chen, Ping Liu, Jing Qian, and Jun Li. 2019. 'Cloud Detection for FY Meteorology Satellite Based on Ensemble Thresholds and Random Forests Approach'. Remote Sensing 11 (1): 44. https://doi.org/10.3390/rs11010044.

Zheng, Caixia, Jesus Pulido, Paul Thorman, and Bernd Hamann. 2015. 'An Improved Method for Object Detection in Astronomical Images'. Monthly Notices of the Royal Astronomical Society 451 (4): 4445–59. https://doi.org/10.1093/mnras/stv1237.

**p10 lines 2-5: "The trajectories.." <– it isn't clear to me what this sentence means. It might need to be rephrased.**

We have rewritten the sentence more clearly and added some additional motivation for the extrapolation of trajectories:

"The feature detection step can often omit the initial or final stages of the evolution of a cloud due to the choice of specific thresholds. Thus, trajectories can also be extrapolated to additional output time steps at the start and at the end of the tracked path. This allows for the inclusion of both the initiation of the cell and the decaying later stages in the analysis of the cloud life cycle."
(page 10, line 21)

**p 10, lines 12-13: "Instead, the algorithm. . . " it might be useful to produce an illustration of this. I'm finding that I'm unable to visualize how cloud trajectories from tobac would appear when there are cloud splits/mergers**

We have extended the description of how the current version of tobac acts in situations of cloud splits/mergers:

"The current implementation of the linking step does not include an explicit treatment of the splitting and merging of clouds, as implemented in several of the cloud tracking algorithms reviewed earlier (Dawe and Austin, 2012; Heus and Seifert, 2013; Heiblum et al., 2016a). Instead, the current version of tobac will create a continuous track with only one of the two separate cloud objects that combine in a merger or evolve from splitting of a tracked object, mostly based on which of these has the more similar direction of travel to the joint object. However, we have structured the implementation of tobac in a way that allows for the future addition of more complex tracking methods that can record a more complex network of relationships between cloud objects at different points in time."
(page 11, line 5)